# Host-directed therapy with amiodarone in preclinical models restricts mycobacterial infection and enhances autophagy

Gül Kilinç,[1] Ralf Boland,[2] Matthias T. Heemskerk,[1] Herman P. Spaink,[2] Mariëlle C. Haks,[1] Michiel van der Vaart,[2] Tom H. M. Ottenhoff,[1] Annemarie H. Meijer,[2] Anno Saris[1]

**ABSTRACT** *Mycobacterium tuberculosis* (*Mtb*) as well as nontuberculous mycobacteria are intracellular pathogens whose treatment is extensive and increasingly impaired due to the rise of mycobacterial drug resistance. The loss of antibiotic efficacy has raised interest in the identification of host-directed therapeutics (HDT) to develop novel treatment strategies for mycobacterial infections. In this study, we identified amiodarone as a potential HDT candidate that inhibited both intracellular *Mtb* and *Mycobacterium avium* in primary human macrophages without directly impairing bacterial growth, thereby confirming that amiodarone acts in a host-mediated manner. Moreover, amiodarone induced the formation of (auto)phagosomes and enhanced autophagic targeting of mycobacteria in macrophages. The induction of autophagy by amiodarone is likely due to enhanced transcriptional regulation, as the nuclear intensity of the transcription factor EB, the master regulator of autophagy and lysosomal biogenesis, was strongly increased. Furthermore, blocking lysosomal degradation with bafilomycin impaired the host-beneficial effect of amiodarone. Finally, amiodarone induced autophagy and reduced bacterial burden in a zebrafish embryo model of tuberculosis, thereby confirming the HDT activity of amiodarone *in vivo*. In conclusion, we have identified amiodarone as an autophagy-inducing antimycobacterial HDT that improves host control of mycobacterial infections.

**IMPORTANCE** Due to the global rise in antibiotic resistance, there is a strong need for alternative treatment strategies against intracellular bacterial infections, including *Mycobacterium tuberculosis* (*Mtb*) and non-tuberculous mycobacteria. Stimulating host defense mechanisms by host‐directed therapy (HDT) is a promising approach for treating mycobacterial infections. This study identified amiodarone, an antiarrhythmic agent, as a potential HDT candidate that inhibits the survival of *Mtb* and *Mycobacterium avium* in primary human macrophages. The antimycobacterial effect of amiodarone was confirmed in an *in vivo* tuberculosis model based on *Mycobacterium marinum* infection of zebrafish embryos. Furthermore, amiodarone induced autophagy and inhibition of the autophagic flux effectively impaired the host-protective effect of amiodarone, supporting that activation of the host (auto)phagolysosomal pathway is essential for the mechanism of action of amiodarone. In conclusion, we have identified amiodarone as an autophagy-inducing HDT that improves host control of a wide range of mycobacteria.

**KEYWORDS** host-directed therapy, amiodarone, *Mycobacterium tuberculosis*, *Mycobacterium avium*, *Mycobacterium marinum*, human macrophages, zebrafish

I n 2022, *Mycobacterium tuberculosis* (*Mtb*) infection affected an estimated 10.6 million people with tuberculosis (TB), of whom 1.3 million died, making TB one of the top 10 leading causes of death globally (1). TB is difficult to treat with classical antibiotics due to the presence of metabolically inactive, i.e., dormant, bacteria inside TB granulomas,

Address correspondence to Annemarie H. Meijer, a.h.meijer@biology.leidenuniv.nl, or Anno Saris, a.saris@lumc.nl.

Gül Kilinç, Ralf Boland, Annemarie H. Meijer, and Anno Saris contributed equally to this article. Author order between them was determined on the basis of involvement in drafting the manuscript.

The authors declare no conflict of interest.

See the funding table on p. 18.

the pathological hallmark of TB (2). These dormant bacteria are far less susceptible to antibiotics (3, 4). The occurrence of multidrug-resistant (MDR) and extensively drug-resistant (XDR) *Mtb* strains further complicates the treatment of TB. While the number of TB cases has been slowly declining in the last decades, a trend that may well be broken as a result of the COVID-19 pandemic (5), the prevalence of infections caused by nontuberculous mycobacteria (NTM) is increasing (1, 6, 7). NTM represent a group of opportunistic mycobacterial pathogens that mostly cause pulmonary diseases (PD), predominantly in populations vulnerable due to immunodeficiencies and/or pre-existing lung conditions. *Mycobacterium avium* (*Mav*) complex accounts for over 80% of the reported NTM-PD cases (8). Despite extensive antibiotic regimens of at least 12 months after negative sputum culture conversion, clinical outcome is poor. Furthermore, *Mav* and several other NTM species display a high level of natural resistance to antibiotics (9). Thus, both for TB and NTM diseases, the development of novel treatment modalities is highly desired.

A promising alternative or adjunctive therapy for mycobacterial infection is host-directed therapy (HDT) (10–14). HDT promotes the host's ability to eliminate invading pathogens either by stimulating host defense mechanisms or alleviating pathogen-induced manipulations of host cellular functions. By targeting host cells, HDT offers several advantages compared to conventional antibiotics: (i) HDT is less likely to result in drug resistance as the pathogen is not directly targeted; (ii) HDT is also effective against MDR/XDR mycobacteria that are insensitive to current standard antibiotics; (iii) HDT has the potential to be effective against dormant bacteria; and (iv) as HDT and antibiotics target different processes, they are expected to act synergistically, which could significantly reduce antibiotic treatment duration and/or dosage, thereby increasing compliance and reducing toxicity. To identify and develop HDT for mycobacterial infection, it is important to understand the host-pathogen interactions (11).

Mycobacteria are predominantly intracellular pathogens and macrophages are the main innate immune cell type wherein they survive and replicate. Macrophages attempt to eliminate mycobacteria in a process whereby mycobacteria are internalized and introduced in mycobacteria-containing phagosomes that mature and ultimately fuse with lysosomes (11, 15, 16). This process should result in the degradation of the content of the formed phagolysosomes by lysosomal hydrolytic enzymes (17). However, mycobacteria are well known for their capability to modulate signaling pathways to escape from host‑defense mechanisms: both *Mtb* and *Mav* can arrest phagosome maturation and potentially escape into the cytosol (11, 17–19). Host cells try to capture and subsequently degrade cytosolic bacteria using the autophagy pathway (20, 21). Studies have already shown that induction of (non)-canonical autophagy in *Mtb*- and *Mav*-infected macrophages restricts intracellular bacterial growth, which supports further research into autophagy as a potential target for HDT (20, 22–24).

A previous drug repurposing screen of a library composed of autophagy-modulating compounds revealed that several antipsychotic drugs as well as the antiarrhythmic drug amiodarone reduce the bacterial burden of *Mtb* in a human cell line (25, 26). Amiodarone functions by blocking calcium, sodium, and potassium channels as well as inhibiting alpha- and beta-adrenergic receptors. Furthermore, amiodarone has been shown to induce autophagy (27–31), and by accumulating in acidic organelles, amiodarone may also interact with other intracellular degradation processes, like the endocytic pathway (32). Whether amiodarone improves host control of mycobacteria, however, has not been established. Here, we aimed to assess the efficacy of amiodarone in reducing mycobacterial burden, both in primary cells and *in vivo,* and to elucidate via which mechanism amiodarone acts as an HDT. To do so, both classically activated pro-inflammatory (M1) macrophages and alternatively activated anti-inflammatory (M2) macrophages were used as surrogates for the polar ends of the human macrophage differentiation spectrum *in vivo* (33). Furthermore, we used the zebrafish (*Danio rerio*) embryo model for TB, in which zebrafish embryos are infected with their natural pathogen *Mycobacterium marinum* (*Mmar*) (34–37), an NTM that shares major virulence factors with *Mtb* and

is frequently used as a surrogate model for TB (37–40). The formation of granulomatous aggregates of leukocytes is recapitulated in the zebrafish TB model (2, 37, 38, 41). Moreover, the zebrafish model has been used to study the role of autophagy in mycobacterial infection, showing that autophagy contributes to host defense *in vivo* (40, 42–44). This makes the zebrafish embryo model for TB a highly suitable model to investigate the role of autophagy in the antimycobacterial effect of amiodarone.

In this study, we aimed to investigate amiodarone as HDT against multiple mycobacterial species in primary human macrophages. Moreover, to understand the mechanism of action of amiodarone, we evaluated the effect of amiodarone on autophagy and the role of autophagy during infection control by amiodarone. Finally, we assessed the efficacy of amiodarone in a zebrafish TB model to determine the *in vivo* translatability.

## RESULTS

### *In vitro* identification of amiodarone as a novel HDT against intracellular mycobacteria

To identify new drugs with host-directed therapeutic activity against intracellular *Mtb*, we have previously screened the Screen-Well autophagy library of clinically approved molecules by treating *Mtb*-infected human cells for 24 hours (25). A promising candidate from this screen was amiodarone (Fig. 1A). To validate the antimycobacterial effect of amiodarone in a physiologically more relevant model, we used a primary human macrophage infection model (26). Classical colony-forming unit (CFU) assays were used to determine the reduction of intracellular *Mtb* load after 24 hours of treatment with 10 µM amiodarone. Amiodarone treatment significantly impaired intracellular bacterial survival in both M1 and M2 macrophages (Fig. 1B; Fig. S1A). To exclude direct antibacterial effects, *Mtb* in liquid broth was exposed to amiodarone at the same concentration, which did not show any effect of amiodarone (Fig. 1C), thereby confirming amiodarone acts in a host-directed manner during *Mtb* infection.

To determine whether amiodarone is exclusive for *Mtb* or may also act against other intracellular pathogenic mycobacteria, the activity of amiodarone was also tested in our M1 and M2 macrophage model infected with the NTM *Mycobacterium avium* (45). Intracellular bacterial survival in primary macrophages, as determined by the mycobacteria growth indicator tube (MGIT) assay, was impaired after amiodarone treatment (Fig. 1D; Fig. S1B). To confirm amiodarone's HDT activity against *Mav*, bacteria were treated with amiodarone in the absence of macrophages. No direct inhibition of bacterial growth was observed (Fig. 1E). Due to our experimental setup, i.e., gentamicin-protection assay, macrophage cell death also results in reduced intracellular bacterial burden. To exclude the involvement of such false-positive results, the effect of amiodarone on cellular viability was determined using the lactate dehydrogenase (LDH) release assay in *Mav*-infected macrophages. There were no indications that amiodarone affected the viability of *Mav*-infected macrophages (Fig. 1F). Taken together, amiodarone was identified as a potential HDT candidate, which impaired the survival of both *Mtb* and *Mav* in primary human pro-inflammatory and anti-inflammatory macrophages.

### Amiodarone enhances the autophagy response to mycobacterial infection

Since amiodarone is known to induce autophagy (28, 29), we assessed this as the potential mechanism of action against mycobacteria. Considering that amiodarone showed the most consistent effect in *Mav*- versus *Mtb*-infected macrophages (standard deviation of 20.1 compared to 28.7, respectively) and that M2 macrophages better resemble alveolar macrophages, which are the primary cells involved during mycobacterial infections (33, 46), we focused on *Mav*-infected M2 macrophages. First, we measured the effect of amiodarone on total protein levels of LC3-II, which is the lipidated form of LC3 that is attached to the (auto)phagosome membrane. These experiments were performed both in the absence and presence of bafilomycin A1 (Baf), a vacuolar-type ATPase inhibitor that impairs lysosomal acidification and thereby blocks the degradation

**A**

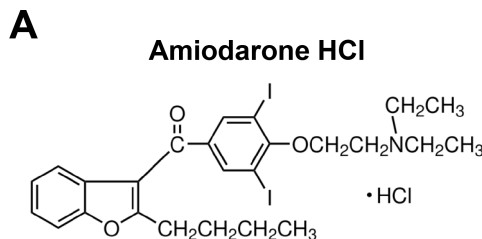

**Amiodarone HCl**

**B**

*Mtb*-M1

*Mtb*-M2

**C**

**D**

*Mav*-M1

*Mav*-M2

**E**

**F**

*Mav*-M1

*Mav*-M2

**FIG 1** Identification of amiodarone as host-directed therapeutic for mycobacterial infections in primary human macrophages. (A) Chemical structure of amiodarone HCl (AMD). (B) *Mtb* H37Rv-infected M1 and M2 macrophages were treated for 24 hours with 10 µM amiodarone or an equal volume of vehicle control dimethyl sulfoxide (DMSO). Cells were subsequently lysed and bacterial survival was determined by CFU assay. Bacterial survival data represent the mean

**FIG 1 (Continued)**

± standard deviation (SD) from different donors ($n$ = 9 or 10). Dots represent the mean from triplicate wells of a single donor. Bacterial survival is expressed as the percentage of vehicle control DMSO (=100%, indicated with the dotted line) per donor. Statistical significance was tested using a paired $t$-test. (C) Growth of *Mtb* H37Rv in liquid broth was monitored for 10 days after exposure to positive control 20 µg/mL rifampicin (RIF), 10 µM amiodarone, or vehicle control DMSO. Data represent the mean ± SD of triplicate wells from three independent experiments. (D) Bacterial survival of *Mav* within M1 and M2 macrophages after treatment for 24 hours with 10 µM amiodarone or an equal volume of vehicle control DMSO. Cells were subsequently lysed and bacterial survival was determined by mycobacteria growth indicator tube (MGIT) assay. Data represent the mean ± SD from different donors ($n$ = 11 or 12). Dots represent the mean from triplicate wells of a single donor. Bacterial survival is expressed as the percentage of vehicle control DMSO (=100%, indicated with the dotted line) per donor. Statistical significance was tested using a paired $t$-test. (E) Growth of *Mav* in liquid broth was monitored for 10 days after exposure to positive control 100 µg/mL kanamycin (KANA), 10 µM amiodarone, or vehicle control DMSO. Data represent the mean ± SD of triplicate wells from three independent experiments. (F) Percentage of viable M1 and M2 macrophages [based on lactate dehydrogenase (LDH) release] after 24 hours of treatment with 10 µM amiodarone or an equal volume of vehicle control DMSO (0.1%, vol/vol). Data represent the mean ± SD from different donors ($n$ = 2). *$P$ < 0.05, **$P$ < 0.01, and ****$P$ < 0.0001.

of (auto)phagosomes, allowing the quantification of total autophagic flux. Amiodarone treatment significantly increased LC3-II protein levels in *Mav*-infected macrophages (Fig. 2A and B; Fig. S2), which persisted in the presence of bafilomycin, indicating that amiodarone promotes both the formation of (auto)phagosomes and the autophagic flux. To further investigate the induction of (auto)phagosomes, we assessed the area of LC3-II puncta in macrophages infected with Wasabi-expressing *Mav* using confocal microscopy (Fig. 2D). Amiodarone treatment resulted in increased LC3-II area in infected macrophages (Fig. 2E). Additionally, colocalization of bacteria and LC3-II-positive vesicles was determined, which showed that amiodarone treatment increased the percentage of bacteria localized in (auto)phagosomes (Fig. 2F). Thus, amiodarone promotes (auto)phagosome formation and flux, which results in enhanced targeting of bacteria to autophagic compartments.

The autophagy response to intracellular pathogens often occurs as a receptor-mediated process (selective autophagy or xenophagy). Therefore, we examined p62, which acts as a cargo receptor that targets ubiquitinated cytoplasmic material (including intracellular bacteria) to (auto)phagosomes for degradation (47, 48). Quantification of total p62-protein levels by western blot showed that treatment of *Mav*-infected macrophages with amiodarone alone did not significantly affect p62 levels (Fig. 2C). In the presence of bafilomycin, a mild, albeit not statistically significant, accumulation of p62 was induced. Moreover, the p62 area and colocalization of Wasabi-expressing *Mav* with p62-positive puncta showed no major alterations upon treatment with amiodarone (Fig. 2G through I). These results suggest that amiodarone might stimulate non-selective (bulk) autophagy, as also occurs during starvation, and the effect on selective autophagy, or a non-canonical autophagy process, remains inconclusive (49).

## Amiodarone increases the activation of the major autophagy regulator TFEB and requires autophagy to eliminate intracellular bacteria

To further investigate the observed effects of amiodarone on infection control and autophagy induction, we focused on transcription factor EB (TFEB), a master regulator of autophagy and lysosomal biogenesis (30, 50–53). Once activated, TFEB enters the nucleus and promotes the expression of autophagy-related genes as well as the coordinated lysosomal expression and regulation gene network (52, 54). Therefore, nuclear intensity of TFEB was assessed in *Mav*-infected M2 macrophages after amiodarone treatment. Compared to untreated controls, a significant increase in nuclear intensity of TFEB was observed in amiodarone-treated cells (Fig. 2J and K).

To establish whether the enhanced autophagic response after amiodarone treatment is required for the reduction of intracellular bacteria, autophagic flux was blocked using bafilomycin in amiodarone-treated *Mav*-infected M2 macrophages. Amiodarone treatment clearly reduced bacterial loads compared to untreated control, but this phenotype was abrogated after blocking (auto)phagosomal lysosomal degradation with bafilomycin (Fig. 2L). Taken together, although we cannot discriminate between bacterial

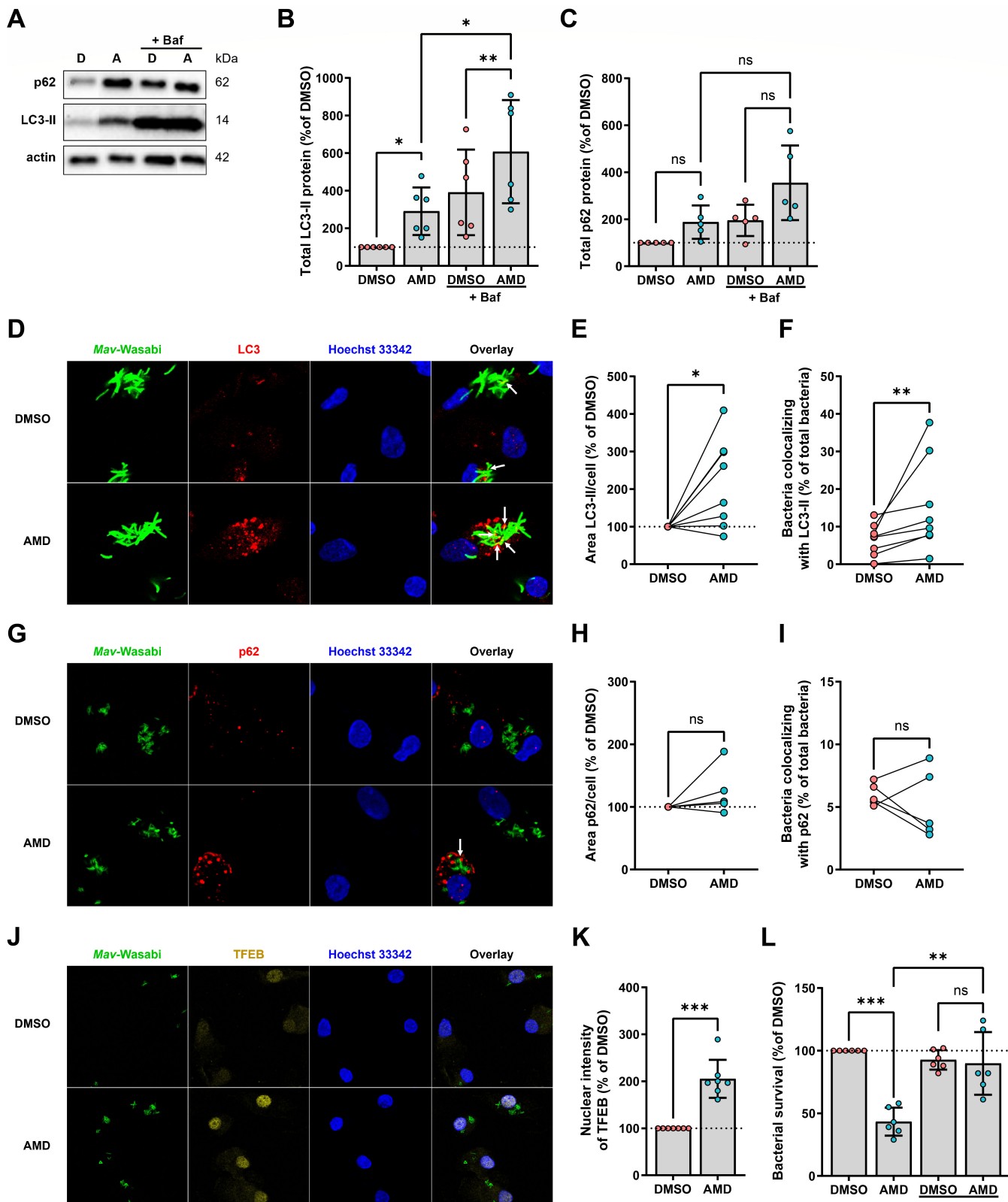

**FIG 2** Amiodarone controls *Mav* infection in primary human macrophages by promoting antimycobacterial autophagy and activating master autophagy regulator TFEB. (A) Western blot analysis of autophagy markers in M2 macrophages treated for 24 hours with 10 µM amiodarone or an equal volume of vehicle control dimethyl sulfoxide (DMSO) (0.1%, vol/vol) in the presence or absence of bafilomycin A1 (Baf) (10 nM) during *Mav* infection. Shown are blots from one

**FIG 2** (Continued)

representative donor out of six donors tested. The image depicts the boxed lanes from the unprocessed original images (Fig. S2). (B) Quantification of LC3-II (+Baf) protein levels from panel A. Protein levels were first normalized to actin and subsequently compared to DMSO control (=100%, indicated with the dotted line) per donor. Data represent the mean ± SD from different donors ($n$ = 6). Statistical significance was tested using a repeated-measures one-way ANOVA with Bonferroni's multiple comparison correction. (C) Quantification of p62 (+Baf) protein levels from panel A. Protein levels were first normalized to actin and subsequently compared to DMSO control (=100%, indicated with the dotted line) per donor. Data represent the mean ± SD from different donors ($n$ = 5). Statistical significance was tested using a repeated-measures one-way ANOVA with Bonferroni's multiple comparison correction. (D) M2 macrophages were treated for 24 hours with 10 µM amiodarone or an equal volume of vehicle control DMSO (0.1%, vol/vol) after infection with Wasabi-expressing *Mav* (green). Cells were subsequently stained with LC3-II (red) and Hoechst 33342 (blue) and analyzed by confocal microscopy. Images shown are of one representative donor out of eight donors tested. Arrows indicate colocalization of *Mav*-Wasabi with LC3-II puncta. (E) Quantification of the LC3-II area per cell count. Dots represent the mean from three wells (three images/well) per condition of a single donor ($n$ = 8). Data are expressed as the percentage of vehicle control DMSO (=100%, indicated with the dotted line) per donor. Statistical significance was tested using a Wilcoxon matched-pairs test. (F) The percentage colocalization (indicated by white arrows in panel D) of intracellular mycobacteria with LC3-II puncta was determined. Dots represent the mean from three wells (three images/well) per condition of a single donor ($n$ = 8). Statistical significance was tested using a Wilcoxon matched-pairs test. (G) M2 macrophages were treated for 4 hours with 10 µM amiodarone or an equal volume of vehicle control DMSO (0.1%, vol/vol) in the presence or absence of 10 nM Baf after infection with Wasabi-expressing *Mav* (green). Cells were subsequently stained with p62 (red) and Hoechst 33342 (blue) and analyzed by confocal microscopy. Images shown are of one representative donor out of five donors tested. Arrows indicate colocalization of *Mav*-Wasabi with p62. (H) Quantification of the p62 area per cell count. Dots represent the mean from three wells (three images/well) per condition of a single donor ($n$ = 5). Data are expressed as the percentage of vehicle control DMSO (=100%, indicated with the dotted line) per donor. Statistical significance was tested using a Wilcoxon matched-pairs test. (I) The percentage colocalization (indicated by white arrows in panel G) of intracellular mycobacteria with p62 was determined. Dots represent the mean from three wells (three images/well) per condition of a single donor ($n$ = 5). Data are expressed as the percentage of vehicle control DMSO. Statistical significance was tested using a Wilcoxon matched-pairs test. (J) Confocal microscopy of Wasabi-expressing (green) *Mav*-infected M2 macrophages treated with 10 µM amiodarone or an equal volume of vehicle control DMSO for 4 hours. Cells were subsequently stained for TFEB (yellow) and Hoechst 33342 (blue). Shown are images of one representative donor out of seven donors tested. (K) Quantification of the total intensity of TFEB within the mark of the cell nucleus. Data represent the mean ± SD from different donors ($n$ = 7). Dots represent the mean from three wells (three images/well) per condition of a single donor. Data are expressed as the percentage of vehicle control DMSO (=100%, indicated with the dotted line) per donor. Statistical significance was tested using a paired *t*-test. (L) Bacterial survival of *Mav* within M2 macrophages after treatment for 24 hours with 10 µM amiodarone or an equal volume of vehicle control DMSO in the absence or presence of 10 nM Baf. Cells were subsequently lysed and bacterial survival was determined by MGIT assay. Data represent the mean ± SD from different donors ($n$ = 6). Dots represent the mean from triplicate wells of a single donor. Bacterial survival is expressed as the percentage of vehicle control DMSO (=100%, indicated with the dotted line) per donor. Statistical significance was tested using a repeated-measures one-way ANOVA with Bonferroni's multiple comparison correction. ns, non-significant; *$P$ < 0.05; **$P$ < 0.01; and ***$P$ < 0.001.

killing or restriction of replication, host-directed therapy with amiodarone controls *Mav* infection in primary human macrophages by promoting antimycobacterial autophagy, which correlates with activation of the master transcriptional regulator TFEB.

## Amiodarone reduces bacterial burden *in vivo*

To validate the activity of amiodarone *in vivo*, we used the zebrafish embryo TB model based on infection with *Mycobacterium marinum*. Before employing this model, we wished to exclude that amiodarone might affect the development or migration properties of zebrafish leukocytes, which would confound the results of infection experiments. Therefore, we used an established injury-based migration assay, the tail amputation assay (55, 56), in a double transgenic neutrophil and macrophage marker line. No alterations in the numbers of neutrophils and macrophages that accumulated at the site of inflammation were observed after treatment with 5 µM amiodarone (Fig. S3). Therefore, we proceeded to assess the effect of amiodarone on infection. Zebrafish embryos were infected 1-day post-fertilization (dpf) with Wasabi-expressing *Mmar*, and treatment was initiated 1-hour post-infection (hpi) with amiodarone in increasing doses (5, 10, and 20 µM). At 4 days post-infection (dpi), the bacterial burden was determined by quantifying the bacterial fluorescent signal using confocal microscopy (Fig. 3A). Amiodarone reduced bacterial burden in a dose-dependent manner at 5 and 10 µM (Fig. 3B; Fig. S4A), without showing any signs of toxicity in zebrafish embryos. The highest dose tested (i.e., 20 µM) induced developmental toxicity (e.g., edema and lethality), and bacterial loads were therefore not quantified. When tested on *in vitro* bacterial cultures, 5 µM amiodarone did not affect *Mmar* growth, while the growth of cultures exposed for

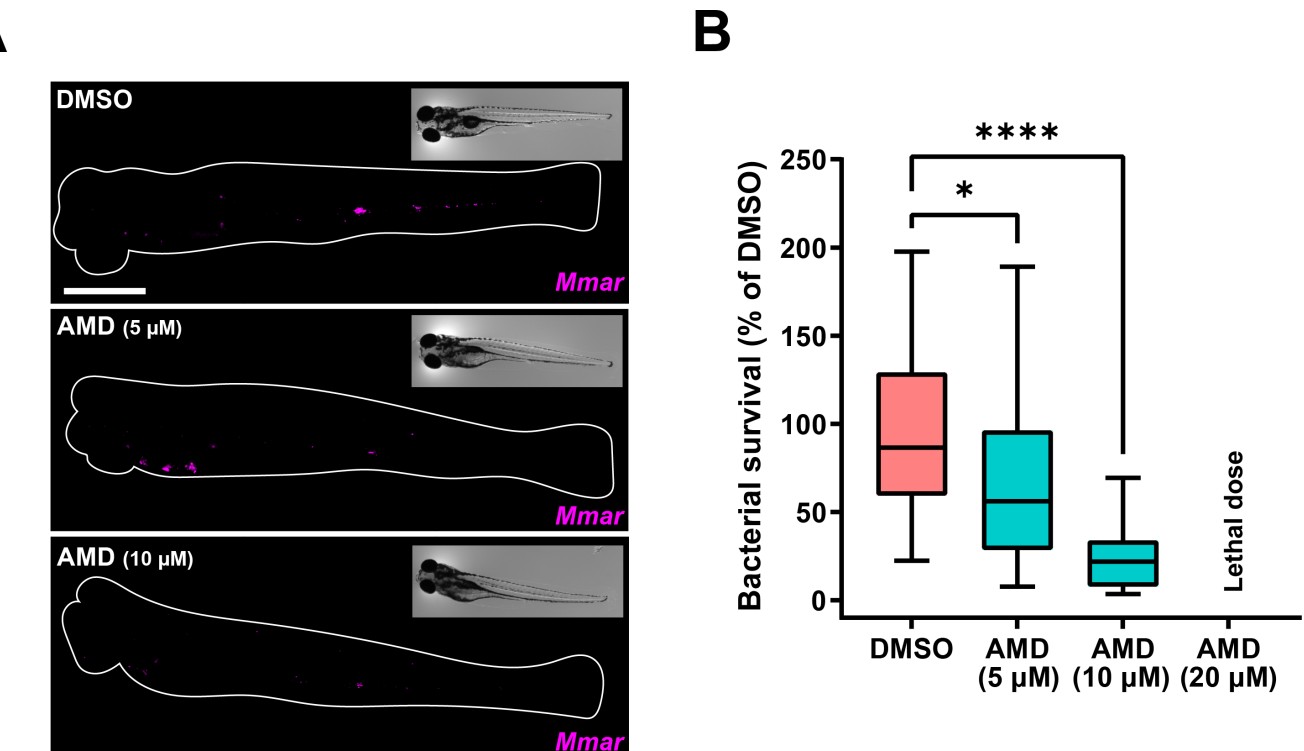

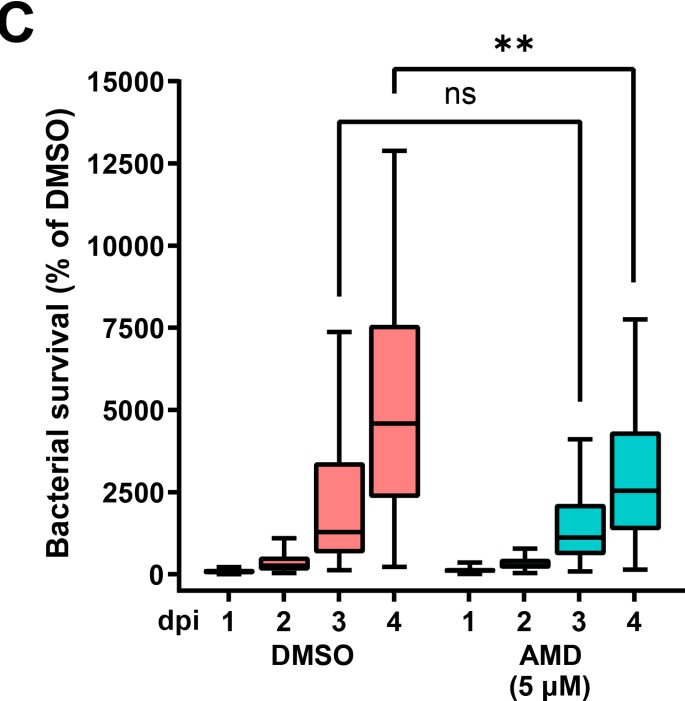

**FIG 3** Amiodarone restricts *Mmar* infection in a host-directed manner. (A) Bacterial burden assay of mWasabi-expressing *Mmar*-infected zebrafish larvae treated with increasing doses of amiodarone (5, 10, and 20 µM) or vehicle control dimethyl sulfoxide (DMSO). Treatment was started at 1 hpi. and larvae were anesthetized at 4 dpi for imaging. Representative stereo fluorescent images of whole larvae infected with mWasabi-expressing *Mmar*. Magenta shows *Mmar*. Scale bar annotates 1 mm. (B) Quantification of bacterial burden shown in panel A. Bacterial burden was normalized to the mean of the control. Data from two independent experiments were combined (*n* = 39–42 per group). Boxplots with 95% confidence intervals are shown, and the black line in the boxplots indicates the group median. Statistical significance was tested using a Kruskal-Wallis with Dunn's multiple comparisons test. (C) Bacterial burden assay of

**FIG 3** (Continued)

mWasabi-expressing *Mmar*-infected zebrafish larvae treated with 5 µM of amiodarone or vehicle control DMSO. Treatment was started at 1 hpi, and larvae were anesthetized at 1, 2, 3, and 4 dpi for imaging. Bacterial burden was normalized to the control (DMSO at 1 dpi), and data from two experimental repeats were combined (*n* = 65–70 per group). Boxplots with 95% confidence intervals are shown, and the black line in the boxplots indicates the group median. Statistical significance was tested using a Kruskal-Wallis with Dunn's multiple comparisons test. ns, non-significant; *$P < 0.05$; **$P < 0.01$; and ****$P < 0.0001$.

48 h to 10 µM amiodarone was inhibited (Fig. S5). Therefore, in subsequent experiments, the dosage of 5 µM amiodarone was used to ensure looking at host-mediated effects. To determine the infection dynamics, bacterial loads were quantified daily from 1 up to 4 dpi. In both the control and treatment groups, bacterial burden increased over time (Fig. 3C; Fig. S4B). Amiodarone treatment, however, significantly impaired the progression of infection, which at 4 dpi resulted in almost a twofold lower bacterial load compared to the control treatment. These results confirm that host-directed therapy with amiodarone reduces mycobacterial loads in a relevant *in vivo* model of TB using zebrafish embryos.

## Amiodarone enhances the formation of (auto)phagosomes *in vivo*

To confirm that the reduced bacterial burden in zebrafish after amiodarone treatment is related to enhanced autophagic activity, as observed in human macrophages, we used a fluorescent zebrafish reporter line for LC3 (GFP-LC3) (57). Embryos at 3 dpf were treated with amiodarone for 24 hours, and GFP-LC3-positive structures were quantified in the tail fin using confocal microscopy (Fig. 4A) (58). Compared to controls, the number of GFP-LC3 structures was significantly increased after amiodarone treatment (Fig. 4B). Subsequently, zebrafish embryos (1 dpf) were infected with mCherry-expressing *Mmar* to investigate whether the increased number of autophagic vesicles after amiodarone treatment colocalized with bacteria. At 2 dpi, embryos were imaged using confocal microscopy in the caudal hematopoietic tissue (CHT) region, the location where infected macrophages are known to aggregate, as an initial step of granuloma formation (38). Both in control and amiodarone-treated embryos, bacterial clusters colocalized with GFP-LC3 clusters, without detectable differences between both groups (Fig. 4C and E). Furthermore, in both groups, an overall increase in the percentage of *Mmar* clusters colocalizing with GFP-LC3 signal was observed, when autophagy flux was blocked with bafilomycin (Fig. 4D and E). In conclusion, while no differences in GFP-LC3-positive *Mmar* clusters were detected, amiodarone showed a marked effect on total GFP-LC3 levels in the zebrafish model, in agreement with our observations in primary human macrophages.

## DISCUSSION

Antibiotic resistance is emerging as one of the principal global health problems for bacterial infections, which impairs the treatment of TB and other difficult-to-treat intracellular bacterial infections, including NTM. Stimulating host defense mechanisms and/or counteracting pathogen‑induced immune modulation by host‑directed therapy is a promising alternative strategy to combat intracellular mycobacterial infections. Here, we report that amiodarone enhances the antimicrobial response of primary human macrophages infected with *Mtb* and *Mav*, paralleled with a significant reduction in mycobacterial burden of *Mmar*-infected zebrafish embryos. Importantly, amiodarone is shown to promote the activity of transcriptional regulator TFEB and induce the formation of (auto)phagosomes and autophagy flux. Inhibition of autophagic flux by blocking lysosomal degradative activity effectively impaired the protective effect of amiodarone, supporting that activation of the host (auto)phagolysosomal pathway is causally involved in the mechanism of action of amiodarone.

This study has identified the host-directed activity of amiodarone both *in vitro* and *in vivo* against both nontuberculous and tuberculosis mycobacteria. Amiodarone is known to induce autophagy and modulate endocytic pathways (28, 29, 32), which may be beneficial during mycobacterial infections as both pathways are crucial processes in the

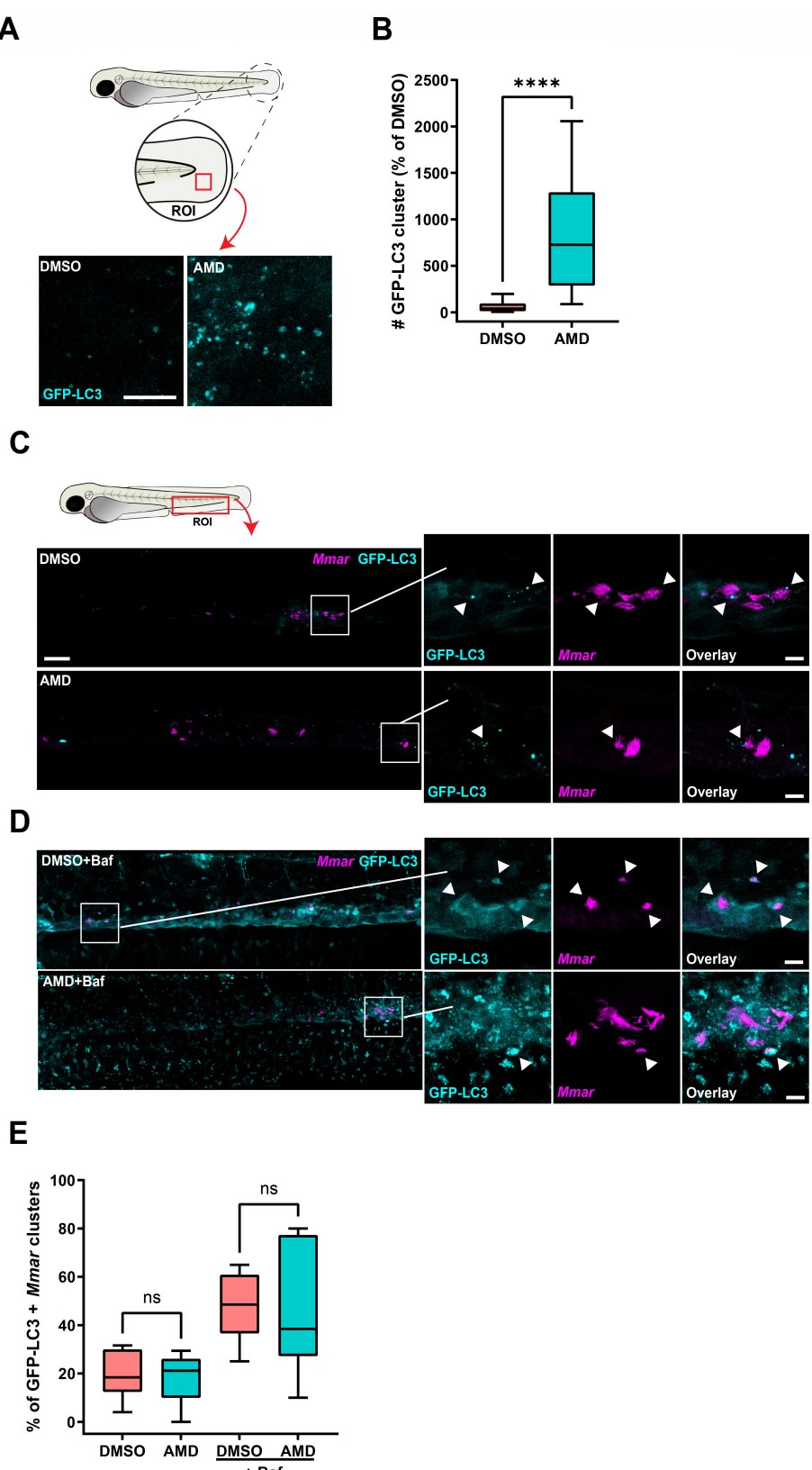

**FIG 4** Amiodarone induces an increase in (auto)phagosomes, without affecting autophagic targeting of *Mmar* clusters. (A) Confocal microscopy max projection of transgenic GFP-LC3 zebrafish larvae treated with 5 µM of amiodarone or vehicle control dimethyl sulfoxide (DMSO). Treatment was started at 3 dpf and larvae were fixed with 4% paraformaldehyde at 4

**FIG 4 (Continued)**

dpf for imaging. Representative max projection images of GFP-LC3-positive vesicles in the indicated region of imaging (ROI) in the tail fin are shown. Cyan shows GFP-LC3-positive vesicles. Scale bar annotates 10 µm. (B) Quantification of GFP-LC3 structures is shown in panel A. Data were normalized to the control, and data from two independent experiments were combined ($n$ = 16–17 per group). Boxplots with 95% confidence intervals are shown, and the black line in the boxplots indicates the group median. Statistical significance was tested using a Mann-Whitney test. (C) Confocal microscopy max projection of mCherry-expressing *Mmar*-infected transgenic GFP-LC3 zebrafish larvae treated with 5 µM of amiodarone or vehicle control DMSO. Treatment was started at 1 hpi, and at 2 dpi, larvae were fixed with 4% paraformaldehyde for imaging. Representative max projection images of the ROI in the CHT region are shown. Cyan shows GFP-LC3-positive vesicles and magenta shows *Mmar*. Scale bar annotates 50 µm. Enlargement of areas indicated in panel C: cyan shows GFP-LC3-positive vesicles and magenta shows *Mmar*. Arrowheads indicate GFP-LC3-positive *Mmar* clusters. Scale bar in the left panel annotates 50 µm and in the right panel 10 µm. (D) Confocal microscopy max projection of mCherry-expressing *Mmar*-infected transgenic GFP-Lc3 zebrafish larvae treated with 5 µM of amiodarone and 160 nm of bafilomycin or vehicle control DMSO. Treatment was started at 1 hpi, and at 2 dpi, larvae were fixed with 4% paraformaldehyde for imaging. Representative max projection images of the ROI in the CHT region are shown. Cyan shows GFP-Lc3-positive vesicles and magenta shows *Mmar*. Scale bar annotates 50 µm. Enlargement of areas indicated in panel D: cyan shows GFP-LC3-positive vesicles and magenta shows *Mmar*. Arrowheads indicate GFP-LC3-positive *Mmar* clusters. Scale bar in the left panel annotates 50 µm and in the right panel 10 µm. (E) Quantification of GFP-LC3-positive *Mmar* clusters in the CHT region shown in panels A and D normalized to the control ($n$ = 8 per group). Boxplots with 95% confidence intervals are shown, and the black line in the boxplots indicates the group median. Statistical analysis was performed using a Kruskal-Wallis with Dunn's multiple comparisons test. ns, non-significant and ****$P < 0.0001$.

intracellular defense against infections with *Mtb* and *Mmar* (20, 40, 42, 59), and has furthermore been suggested for *Mav* (60, 61). Amiodarone increased autophagic flux both *in vitro* in primary human macrophages and *in vivo*. In primary human macrophages, we could additionally demonstrate that amiodarone promoted autophagic targeting of intracellular *Mav*. Targeting bacteria to autophagosomes for degradation is a specific form of canonical autophagy called xenophagy (48). LC3, however, is not a specific autophagosome marker since LC3 can also be lipidated to phagosomes in a non-canonical autophagy process such as LC3-associated phagocytosis (62), which uses components of the canonical autophagy machinery but selects cargo extracellularly (62). Conversely, bacterial control during xenophagy is dependent on the ability of bacteria to escape the phagosome into the cytosol. Phagosomal escape is known as one of the virulence mechanisms of *Mtb* and *Mmar* (17–19, 63, 64), but whether *Mav* also escapes phagosomes remains unknown. Amiodarone treatment of *Mav*-infected macrophages did not significantly modify the levels of p62 (e.g., SQSTM1), one of the most well-known receptors targeting ubiquitinated cytosolic cargo to the autophagosome. Moreover, colocalization of bacteria with p62 was not increased upon amiodarone treatment. Although the involvement of other cargo receptors including NDP52 and Optineurin cannot be excluded (42, 48, 65), our data are supported by others who showed that amiodarone induced non-canonical autophagy independently of the canonical autophagy pathway (28).

Amiodarone is able to induce autophagy via mTOR-independent and -dependent pathways. Amiodarone can induce mTOR-independent autophagy by blocking calcium-mediated production of calpains (66). Calpains stimulate the production of cAMP, which inhibits autophagy via the cyclical mTOR-independent pathway (67, 68) and are suggested to cleave Atg5, which is required for the formation of autophagosomes (69). When sufficient cellular nutrients are available, mTORC1 inhibits autophagy by interacting directly with ULK1, an important enzyme for the initiation of autophagosome biogenesis (21). In addition, mTORC1 impairs the nuclear translocation and activation of TFEB by promoting its phosphorylation (70–72). When mTOR is inhibited, by starvation or lysosomal dysfunction (e.g., phospholipidosis), dephosphorylated TFEB translocates to the nucleus where it coordinates the transcriptional program to increase lysosomal biogenesis and autophagy (30, 51, 73). Our finding that co-treatment with lysosomal activity inhibitor bafilomycin abrogated the antimycobacterial effect of amiodarone

shows that lysosomal degradation is operational and instrumental for the host-protective effect of amiodarone. Amiodarone furthermore enhanced the nuclear intensity of TFEB in *Mav*-infected macrophages, likely by inhibition of mTOR (66). In agreement, overexpression of TFEB was previously shown to potentiate autophagy (30). By potentially interacting with multiple players from the autophagy machinery, amiodarone might prove to be a robust activator for autophagy in varying conditions, including during mycobacterial infection.

In addition to its effects on autophagy pathways, amiodarone has been reported to impair the function of certain lysosomal enzymes and induce phospholipidosis (30, 74). Phospholipidosis is a phospholipid storage disorder, characterized by the accumulation of phospholipids within lysosomes, which cells try to overcome by promoting autophagy (30, 75). Thus, the effect of amiodarone may depend on the phospholipidosis-mediated induction of autophagy during TFEB overexpression. For SARS-CoV-2, the *in vitro* activity of phospholipidosis-inducing drugs failed to translate *in vivo,* which hampers drug discovery (76). Regardless, amiodarone has been shown to induce phospholipidosis in rodents (77–79). In our study, the host-directed effect of amiodarone was reproducible in zebrafish, suggesting that the activity of drugs that may act by inducing phospholipidosis can be translated *in vivo* for mycobacterial infections. To control mycobacterial infections in our study, amiodarone was used in concentrations of 5 and 10 µM. In patients treated with amiodarone, peak serum concentrations are reported to range from 1 to 5.1 µM and can increase up to 11 µM shortly after intravenous administration (80, 81). Nevertheless, amiodarone has a number of well-known and occasional serious side effects upon chronic usage including lung toxicity, which are more common in patients with plasma levels exceeding 3.9 µM (82–84). As the amiodarone concentrations required to mediate a protective host-directed effect during mycobacterial infections are on the high end of patient serum concentrations, complicating the clinical applicability of amiodarone, identifying autophagy-inducing compounds with a more favorable safety profile is highly desirable to aid clinical translation. For *Mtb*, the relevance of promoting autophagy is currently being investigated in clinical trials (85, 86), and our study underlines that promoting autophagy may also be beneficial in patients infected with nontuberculous mycobacteria.

Our study might have several limitations. First, although we identified the HDT activity of amiodarone against multiple mycobacterial species, the role of autophagy in infection control by amiodarone was only shown during *Mav* infection. If mycobacteria differ in their intracellular behavior and the way they are degraded, our findings regarding autophagy during *Mav* infection might not apply to *Mtb* or *Mmar*. Second, we showed that induction of autophagy by amiodarone is required for infection control and that TFEB activation upon amiodarone treatment is enhanced. However, our study lacks evidence that TFEB activation is promoting the autophagy-mediated HDT activity of amiodarone. Third, we did not evaluate the efficacy of amiodarone in combination with standard-of-care antibiotics to detect any cumulative or synergistic effects. Clinical application of HDT, however, will most likely be considered as an adjunctive therapy to standard of care used to treat mycobacterial infections (11). Despite these limitations, we support the possible clinical applicability of amiodarone by showing that amiodarone improved host control of mycobacterial infections both *in vitro* and *in vivo*. Furthermore, our study presents a new autophagy-inducing compound suitable for drug repurposing. Drug repurposing as HDT has various advantages, including the known safety profile of drugs and the faster facilitation of the identification of HDT to treat mycobacterial infections.

Taken together, amiodarone acts as a host-directed therapeutic in primary human macrophages and in zebrafish against nontuberculous and tuberculous mycobacterial strains. Amiodarone induces autophagy, most likely by promoting the nuclear translocation of TFEB and concomitant upregulation of proteins involved in autophagy, and activation of the (auto)phagolysosomal pathway by amiodarone interferes with the ability of mycobacteria to survive intracellularly. While our study shows the feasibility of

exploiting autophagy as a target for HDT during *Mtb* as well as NTM infections, further understanding of the molecular mechanisms of how autophagy is regulated and controls mycobacterial infection will enable the development of autophagy-modulating HDT with a more favorable therapeutic index.

## MATERIALS AND METHODS

### Reagents and antibodies

Anti-human CD163-PE, CD14-PE-Cy7, and CD1a-Alexa Fluor 647 (1:20) were obtained from Biolegend (Amsterdam, the Netherlands) and anti-human CD11b-BB515 (1:20) from BD Biosciences. For confocal microscopy, the following antibodies were used: rabbit anti-human LC3A/B (1:200) and rabbit anti-human TFEB (1:200) from Cell Signaling Technology (Leiden, the Netherlands), mouse anti-human SQSTM1/p62 (1:200) from Santa Cruz Biotechnology (Heidelberg, Germany), and donkey anti-rabbit IgG (H + L)-Alexa Fluor 555 (1:200) and goat anti-mouse IgG (H + L)-Alexa Fluor 647 (1:200) from Abcam (Amsterdam, The Netherlands). Hoechst 33342 (1:2,000) was purchased from Sigma-Aldrich (Zwijndrecht, the Netherlands). For western blot, rabbit anti-human LC3B (1:500) from Novus Biologicals/Bio-Techne (Abingdon, UK), mouse anti-human SQSTM1/p62 (1:500) from Santa Cruz Biotechnology (Heidelberg, Germany), and mouse anti-human $\beta$-actin (1:1,000) from Sigma-Aldrich were used. Horseradish peroxidase-conjugated goat anti-rabbit IgG (H + L) and goat anti-mouse IgG (H + L) (1:5,000) were purchased from Invitrogen, ThermoFisher Scientific.

Dimethyl sulfoxide (DMSO), amiodarone HCl, bafilomycin A1, rifampicin, and kanamycin sulfate were purchased from Sigma-Aldrich.

### Cell culture

Buffy coats were obtained from healthy anonymous donors (Dutch adults) after written informed consent (Sanquin Blood Bank, Amsterdam, The Netherlands). Primary human macrophages were obtained as described previously (45). In short, CD14+ monocytes were isolated from peripheral blood mononuclear cells by density gradient centrifugation over Ficoll (Pharmacy, LUMC, the Netherlands) and by magnetic-activated cell sorting using anti-CD14-coated microbeads (Miltenyi Biotec, Auburn, CA, USA). Purified CD14+ monocytes were cultured for 6 days at 37°C/5% $CO_2$ in Gibco Dutch modified Roswell Park Memorial Institute (RPMI) 1640 medium (ThermoFisher Scientific, Landsmeer, the Netherlands) supplemented with 10% fetal calf serum, 2 mM L-glutamine (PAA, Linz, Austria), 100 units/nL penicillin, 100 µg/mL streptomycin, and either 5 ng/mL granulocyte-macrophage colony-stimulating factor (ThermoFisher Scientific) or 50 ng/mL macrophage colony-stimulating factor (R&D Systems, Abingdon, UK) to promote pro-inflammatory M1 or anti-inflammatory M2 macrophage differentiation, respectively. Cytokines were refreshed on day 3 of differentiation. One day prior to experimental procedures, macrophages were harvested by trypsinization with 0.05% Trypsin-EDTA (ThermoFisher Scientific) and scraping and seeded into flat-bottom 96-well plates (30,000 cells/well) if not indicated otherwise. The M1 and M2 macrophage differentiation was validated based on cell surface marker expression (CD11b, CD1a, CD14, and CD163) as determined by flow cytometry and quantification of cytokine production (IL-10 and IL-12) using ELISA following 24 hours of stimulation of cells with 100 ng/mL lipopolysaccharide (InvivoGen, San Diego, USA).

### Bacterial cultures

*Mav* laboratory strain 101 (700898, ATCC, VA, USA), *Mtb* (wild-type H37Rv), and mCherry-expressing *Mmar* M-strain were cultured as described previously (25, 45, 87). Bacterial concentrations were determined by measuring the optical density of planktonic cultures at 600 nm ($OD_{600}$).

## Cell-free bacterial growth assay

Mtb, Mav, and Mmar cultures were diluted to an $OD_{600}$ of 0.1 in Difco Middlebrook 7H9 broth (Becton Dickinson, Breda, the Netherlands), containing 0.2% glycerol (Merck Life Science, Amsterdam, the Netherlands), 0.05% Tween-80 (Merck Life Science), 10% Middlebrook albumin, dextrose, and catalase enrichment (Becton Dickinson), and 100 µg/mL Hygromycin B (Life Technologies-Invitrogen, Bleiswijk, the Netherlands), of which 50 µL per flat-bottom 96 well of Mtb and Mav and 5 mL of Mmar were incubated with 50 µL of chemical compounds or DMSO (0.1%, vol/vol) at indicated concentration at 37°C/5% $CO_2$. Bacterial growth of Mtb and Mav was monitored until 10–14 days of incubation, and Mmar growth was measured during 2 days of incubation at 28.5°C. Absorbance at a 600 nm wavelength was measured directly after plating and at indicated time points following resuspension of wells on the Envision Multimode Plate Reader (Perkin Elmer).

## Bacterial infection and treatment of cells

One day prior to infection, Mtb or Mav was diluted to a density corresponding with early log-phase growth ($OD_{600}$ of 0.25) to reach the log phase during infection. On the day of infection, bacterial suspensions were diluted in a cell culture medium without antibiotics to infect macrophages with a multiplicity of infection (MOI) of 10. The accuracy of the MOI was verified by a standard CFU (45).

After the addition of bacteria to the cells, plates were centrifuged for 3 minutes at 130 rcf and incubated for 1 hour at 37°C/5% $CO_2$. Extracellular bacteria were removed, and cells were treated with fresh RPMI 1640 containing 30 µg/mL gentamicin for 10 minutes to eradicate residual extracellular bacteria. Cells were subsequently incubated at 37°C/5% $CO_2$ in RPMI 1640 medium supplemented with 5 µg/mL gentamicin and, if indicated, compounds at indicated concentration or vehicle control (DMSO 0.1%, vol/vol) until readout. Following the treatment of cells, the supernatant was removed, and the cells were either lysed using 100 or 125 µL of lysis buffer ($H_2O$ + 0.05% SDS) for the determination of intracellular bacterial burden using a CFU assay or the MGIT system (45), or processed for western blot or confocal microscopy analysis. The activity of amiodarone on the elimination of bacteria was determined by calculating the fraction of intracellular bacteria measured after treatment compared to the control.

## Lactate dehydrogenase release assay

Cells (30,000 cells/well) were infected and treated as described above and centrifuged for 3 minutes at 130 rcf. Supernatants were transferred to a new plate and reacted with substrate mix from the Cytotoxicity Detection kit (LDH) (Merck Life Science) for 30 minutes at room temperature in the dark. Absorbance at $OD_{485}$ and $OD_{690}$ was measured using an Envision Plate Reader. Toxicity was calculated using the absorbance values and the formula (experimental sample – untreated sample)/(positive control sample – untreated sample), where the positive control indicates cells lysed using 2% Triton X-100 (Sigma-Aldrich). Cell viability was determined as the inverse value of toxicity, where 100% indicates the cell viability of the untreated sample.

## Western blot analysis

Cell lysates (300,000 cells/well in 24-well plates) were prepared and protein concentrations were measured as described previously (88). Cell lysates were loaded on a 15-well 4%–20% Mini-PROTEAN TGX Precast Protein Gel (Bio-Rad Laboratories, Veenendaal, the Netherlands), and Amersham ECL Full-Range Rainbow Molecular Weight Marker (Sigma-Aldrich) was added as a reference. Proteins were transferred to ethanol-activated Immun-Blot PVDF membranes (Bio-Rad) in Tris-glycine buffer (25 mM Tris, 192 mM glycine, and 20% methanol). Subsequently, membranes were blocked for 45 minutes in PBS with 5% non-fat dry milk (PBS/5% milk) (Campina, Amersfoort, The Netherlands) and probed with the indicated antibodies in PBS/5% milk for 90 minutes at RT.

Membranes were washed and incubated two times for 5 minutes with PBS + 0.75% Tween-20 (PBST) and stained with secondary antibodies in PBS/5% milk for 45 minutes at RT. Membranes were washed and incubated two times for 5 minutes with PBST before revelation using enhanced chemiluminescence SuperSignal West Dura extended duration substrate (ThermoFisher Scientific). Imaging was performed on an iBright Imaging System (Invitrogen, Breda, the Netherlands). Protein bands were quantified using ImageJ/Fiji software (NIH, Bethesda, MD, USA) and normalized to actin.

## Confocal microscopy of cells

For confocal microscopy, cells (30,000 cells/well) were cultured in pre-washed poly-d-lysine-coated glass-bottom 96-well plates (no. 1.5, MatTek Corporation, Ashland, MA, USA). Following infection and treatment, cells were fixed with 1% (wt/vol) formaldehyde (ThermoFisher Scientific) for 1 hour at RT, washed twice, and Fc receptors were blocked with 5% human serum diluted in PBS (PBS/5% HS) for 45 minutes at RT. Next, cells were stained with indicated antibodies in PBS/5% HS for 30 minutes at RT, washed twice with PBS/5% HS, and incubated with secondary antibodies for 30 minutes at RT in the dark. Cells were incubated with 5 µg/mL Hoechst 33342 for 10 minutes at RT in the dark and mounted overnight using ProLong Glass Antifade Mountant (Invitrogen, ThermoFisher Scientific, Landsmeer, the Netherlands). Plates were imaged by taking three images per well, using a Leica SP8WLL Confocal microscope (Leica, Amsterdam, the Netherlands) equipped with a 63× oil immersion objective.

Image analysis was performed as follows: LC3 and p62 channels were background subtracted in ImageJ/Fiji software with rolling ball algorithm using a 20-pixel radius (89). CellProfiler 3.0.0. was used for the segmentation of both the fluorescent bacteria and the markers of interest with global manual thresholding (bacteria) and adaptive two or three-class Otsu thresholding (LC3 and p62, respectively) (90). The area of each fluorescent marker was specified for each image and was normalized to cell count based on Hoechst 33342 staining. The percentage of overlap, i.e., colocalization, of *Mav* with LC3 and p62 was calculated for each image, and the average colocalization was determined for each treatment condition. The integrated/mean intensity of TFEB per single nucleus was used to determine the nuclear presence of TFEB.

## Zebrafish culture

Zebrafish lines (Table S1) were maintained according to standard protocols (www.zfin.org). Zebrafish eggs were obtained by natural spawning of single crosses to achieve synchronized developmental timing. Eggs from at least five couples were combined to achieve heterogeneous groups. Eggs and embryos were kept in egg water (60 µg/mL sea salt, Sera Marin, Heinsberg, Germany) at ~28.5°C after harvesting and in embryo medium after infection and/or treatment (E2, buffered medium, composition: 15 mM NaCl, 0.5 mM KCl, 1 mM MgSO4, 150 µM $KH_2PO_4$, 1 mM $CaCl_2$, and 0.7 mM $NaHCO_3$) at ~28.5°C for the duration of experiments.

## Bacterial infection and treatment of zebrafish embryos

Zebrafish embryos were infected with *Mmar* inoculum resuspended in PBS containing 2% (wt/vol) polyvinylpyrrolidone (PVP40). The injection dose was determined by optical density measurement ($OD_{600}$ of 1 corresponds to ~100 CFU/nL). Infection experiments were conducted according to previously described procedures (35, 87). In brief, microinjections were performed using borosilicate glass microcapillary injection needles (Harvard Apparatus, 300038, 1 mm O.D. × 0.78 mm I.D.) prepared using a micropipette puller device (Sutter Instruments Flaming/Brown P-97). Needles were mounted on a micromanipulator (Sutter Instruments MM-33R) positioned under a stereo microscope. Prior to injection, embryos were anesthetized using 200 µg/mL buffered 3-aminobenzoic acid ethyl ester (Tricaine, Sigma-Aldrich) in egg water. They were then positioned on a 1% agarose plate (in egg water) and injected with a 1-nL inoculum

containing ~200 CFU *Mmar* at 30 hours post-fertilization in the blood island or at 3 dpf in the tail fin (58). Treatment of zebrafish embryos was performed by immersion. Stock concentrations were diluted to treatment doses in a complete embryo medium without antibiotics. As a solvent control treatment, DMSO was diluted to the same concentration (%, vol/vol) as amiodarone treatment.

For the assessment of bacterial burden, larvae were anesthetized using tricaine at 4 dpi, positioned on a 1% agarose (in egg water) plate, and imaged using a Leica M205 FA stereo fluorescence microscope equipped with a DFC345 FX monochrome camera. Bacterial burden was determined based on fluorescent pixel quantification (Stoop 2011). For confocal imaging, larvae were either fixed in 4% paraformaldehyde in PBS at 20°C for 2 hours or at 4°C or anesthetized using tricaine and embedded in 1.5% low melting point agarose (in egg water) before imaging (58). Time points of all confocal experiments are described in the figure legends.

## Confocal microscopy of zebrafish

To visualize fixed 4-dpf uninfected or 1-dpi larvae, larvae were embedded in 1.5% low melting point agarose (weight per volume, in egg water) and imaged using a Leica TCS SPE confocal 63× oil immersion objective (HC PL APO CS2, NA 1.42) and a Leica TCS SP8 confocal microscope with a 40× water immersion objective (HCX APO L U-V-I, NA 0.8).

For the visualization of LC3 dynamics, Tg(CMV:EGFP-map1lc3b) larvae were embedded in 1.5% low melting point agarose (weight per volume, in egg water) and imaged using a Leica TCS SPE confocal microscope. Imaging was performed using a 63× oil immersion objective (HC PL APO CS2, NA 1.42) in a region of the tail fin to detect EGFP-map1lc3b, further referred to as GFP-LC3-positive vesicles. To determine colocalization between *Mmar* and GFP-LC3, larvae were embedded in 1.5% low melting agarose (in egg water) and imaged in the caudal hematopoietic tissue, using a Leica TCS SP8 confocal microscope with a 40× water immersion objective (HCX APO L U-V-I, NA 0.8). Images were obtained using Leica Las X software. For the quantification of GFP-LC3 levels, the find maxima algorithm with a noise tolerance of 50 was used in Fiji software version 1.53c. To determine the association of GFP-LC3 with bacteria, manual counting was performed on the obtained confocal images using Leica Las X software.

## Tail amputation assay

Embryos of a Tg(mpeg1:mcherryF)/Tg(mpx:gfp) double transgenic line were anesthetized using tricaine at 3 dpf, positioned on a 1% agarose (in egg water) plate, and the tails were partially amputated with a 1 mm sapphire blade (World Precision Instruments) under a Leica M165C stereomicroscope (91). After amputation, larvae were incubated in an embryo medium for 4 hours and fixed using 4% paraformaldehyde. After fixation, larvae were positioned on a 1% agarose (in egg water) plate and imaged using a Leica M205 FA stereo fluorescence microscope equipped with a DFC345 FX monochrome camera. Macrophages were detected based on the fluorescence of their mCherry label, and neutrophils were detected based on their GFP label. The number of leukocytes recruited to the wounded area was counted as described previously (91).

## Statistical analysis

To evaluate the statistical relevance of observed differences for parametric paired data sets (normal distribution was determined using the Shapiro-Wilk normality test), a paired *t*-test when comparing two groups and repeated measures one-way ANOVA when comparing three or more groups were used. Nonparametric paired data sets were tested with the Wilcoxon matched-pairs test. In case of unpaired samples (i.e., zebrafish experiments), Mann-Whitney test or Kruskal-Wallis with Dunn's multiple comparisons test was applied when assessing the differences between two or more groups, respectively. Data were normalized to the mean of the control group and independent repeats were combined unless otherwise indicated. The number of experiments combined is indicated in the figure legend for each experiment.

Analyses and graphical representation were performed using GraphPad Prism 8.0 and 9.0 (GraphPad Software, San Diego, CA, USA), with *P*-values < 0.05 considered statistically significant.

## ACKNOWLEDGMENTS

We gratefully acknowledge Arthur Eibergen from the Department of Infectious Diseases, Leiden University Medical Center for his contribution to the validation of MGIT as an assay applicable for testing HDT, and Nils Olijhoek, Amy M. Barclay, Daniel C.M. van der Hoeven, and Elisa van der Sar from the Institute of Biology Leiden, Leiden University for their assistance in the experimental work that provided data used in this manuscript. We also gratefully acknowledge Gabriel Forn-Cuní from the Institute of Biology Leiden, Leiden University for the use of his raincloud plots application.

This project has received funding from the Innovative Medicines Initiative 2 Joint Undertaking (IMI2 JU) (www.imi.europa.eu) under the RespiriNTM (grant No. 853932) and the RespiriTB (grant No. 853903) project within the IMI AntiMicrobial Resistance (AMR) Accelerator program (the JU receives support from the European Union's Horizon 2020 research and innovation programme and EFPIA), NWO Domain Applied and Engineering Sciences (NWO-TTW grant 13259). The funders had no role in study design, data collection and analysis, decision to publish, or preparation of the manuscript.

G.K., R.B., M.T.H., H.P.S., M.C.H., M.v.d.V., T.H.M.O., A.H.M., and A.S. designed the experiments. G.K., R.B., and M.T.H. performed the experiments and processed the experimental data. G.K., R.B., M.T.H., M.C.H, T.H.M.O., A.H.M., and A.S. contributed to the interpretation of the results. G.K. and R.B. designed the figures and wrote the original draft. G.K., R.B., M.T.H., M.C.H., M.v.d.V., T.H.M.O., A.H.M., and A.S. contributed to the final review and editing of the manuscript. M.C.H., M.v.d.V., T.H.M.O., A.H.M., and A.S. supervised the project. M.C.H., H.P.S., A.H.M., and T.H.M.O. secured funding. All authors reviewed the manuscript.

## AUTHOR AFFILIATIONS

[1]Department of Infectious Diseases, Leiden University Medical Center, Leiden, the Netherlands
[2]Institute of Biology Leiden, Leiden University, Leiden, the Netherlands

## PRESENT ADDRESS

Ralf Boland, Dutch Research Council (NWO), The Hague, the Netherlands
Matthias T. Heemskerk, Galapagos NV, Oegstgeest, the Netherlands
Michiel van der Vaart, Dutch Heart Foundation, The Hague, the Netherlands

## AUTHOR ORCIDs

Gül Kilinç  http://orcid.org/0000-0003-3521-3842
Ralf Boland  http://orcid.org/0000-0002-1463-5977
Matthias T. Heemskerk  http://orcid.org/0000-0002-7580-4658
Herman P. Spaink  http://orcid.org/0000-0003-4128-9501
Mariëlle C. Haks  http://orcid.org/0000-0002-7538-1800
Michiel van der Vaart  http://orcid.org/0000-0003-0828-7088
Tom H. M. Ottenhoff  http://orcid.org/0000-0003-3706-3403
Annemarie H. Meijer  http://orcid.org/0000-0002-1325-0725
Anno Saris  http://orcid.org/0000-0003-0493-9501

## FUNDING

| Funder | Grant(s) | Author(s) |
|---|---|---|
| Innovative Medicines Initiative 2 Joint Undertaking - RespiriNTM | 853932 | Gül Kilinç |
| Innovative Medicines Initiative 2 Joint Undertaking - RespiriTB | 853903 | Anno Saris |
| NWO Domain Applied and Engineering Sciences | 13259 | Ralf Boland |
| | | Matthias T. Heemskerk |

## ETHICS APPROVAL

This study was approved by the Sanquin Ethical Advisory Board, in accordance with the Declaration of Helsinki and according to Dutch regulations.

Zebrafish were maintained and handled in compliance with the local animal welfare regulations as overseen by the Animal Welfare Body of Leiden University (license number: 10612). All practices involving zebrafish were performed in accordance with European laws, guidelines and policies for animal experimentation, housing, and care (European Directive 2010/63/EU on the protection of animals used for scientific purposes). The present study did not involve any procedures within the meaning of Article 3 of Directive 2010/63/EU and as such is not subject to authorization by an ethics committee.

## ADDITIONAL FILES

The following material is available online.

### Supplemental Material

**Supplemental material (Spectrum00167-24-s0001.docx).** Fig. S1 to S5; Table S1.

### Open Peer Review

**PEER REVIEW HISTORY (review-history.pdf).** An accounting of the reviewer comments and feedback.

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
