## [Reviewer comments · Microbiology Spectrum]

Microbiology Spectrum

Host-directed therapy with amiodarone in preclinical models restricts mycobacterial infection and enhances autophagy

Gül Kiling, Ralf Boland, Matthias Heemskerk, Herman Spaink, Mariëlle Haks, Michiel van der Vaart, Tom H. Ottenhoff, Annemarie Meijer, and Anno Saris

Corresponding Author(s): Anno Saris, Leids Universitair Medisch Centrum

Review Timeline:

Submission Date:	February 28, 2024
Editorial Decision:	April 8, 2024
Revision Received:	May 21, 2024
Accepted:	May 23, 2024

Editor: Selvakumar Subbian

Reviewer(s): The reviewers have opted to remain anonymous.

Transaction Report:

DOI: <https://doi.org/10.1128/spectrum.00167-24>

Re: Spectrum00167-24 (Host-directed therapy with amiodarone restricts mycobacterial infection and enhances autophagy)

Dear Dr. Anno Saris:

Thank you for submitting your work for publication in Spectrum. Below you will find my comments, instructions from the Spectrum editorial office, and the reviewer comments.

Please incorporate all your comments/response to these queries in the revised manuscript, in addition to the rebuttal letter.

Editor comments:

Figure-1B, 1D and 1F- please provide the actual raw values/numbers for each group in these figures as supplementary data.

Figure 2A- please provide the raw/full length gel picture(s) as supplementary data.

Figure 2D and G- the white arrows are barely visible; it would look better/clear if the size can be increased.

Figure-3B, and 3C- please provide the actual raw values/numbers for each group in these figures as supplementary data.

Figure-4 has 2 panels labelled as C and none with D. Kindly correct this error.

Line 208. It is important to disclose if the highest dose of AMD induced any developmental toxicity, even if it was not quantified?

Lines 94-96. It appears that Ref#24 reports on antipsychotic drugs and no mentioning of amiodarone. Clarify and revise.

Line 134-135. This is confusing, since, indeed, AMD showed a direct antimycobacterial effect on *M. marinum* as per this paper. Kindly revise this sentence and clarify the differential responses between *Mtb*, *Mav* and *Mm* regarding AMD activity.

Line 153. It is important to explain what is the rationale go after M2 macrophages and not M1 for subsequent assays?, and what is the impact of AMD on autophagy in M1 cells with or without mycobacterial infection?

Based on this report, it appears that *Mm* and *Mtb* are behaving differently in response to different doses of AMD (compare Figures 1C, 1E and Suppl.Fig1A). Similarly, *Mtb* and *Mav* infected macrophages differ in the way they were handled through autophagic pathways (lines 261-270). However, the data shown in Figure 2 pertains only to *Mav*-infected macrophages and not *Mtb*-infected counterpart. Since these are major weaknesses, it is important to include a section on the limitations of this study, particularly the different model systems used, and different doses of AMD used in these models and lack of direct mechanistic link between the targeted pathways etc., at the end of discussion section.

Revision Guidelines

Sincerely,
Selvakumar Subbian
Editor
Microbiology Spectrum

Reviewer #1 (Comments for the Author):

Summary:

The authors identified amiodarone as a potential HDT candidate that inhibited both intracellular Mtb and Mycobacterium avium (Mav) in primary human macrophages, without directly impairing bacterial growth, thereby confirming that amiodarone acts in a host-mediated manner. They demonstrate that amiodarone induced autophagy and reduced bacterial burden in a zebrafish embryo model of TB, thereby confirming the HDT activity of amiodarone in vivo. They conclude that amiodarone is an autophagy-inducing anti-mycobacterial HDT that improves host control of mycobacterial infections.

Comment 1: Please use the latest WHO Global TB report when reporting TB figures.

Comment 2: Lines 114-121: this section appears at the end of the introduction, typically where the authors would state the aim, objective, hypothesis, rationale for the research, but this section reads rather like a summary or conclusion that one would find at the end of a manuscript. I would recommend that the authors rewrite this section removing any comments on findings or conclusions made.

Comment 3: The title is slightly misleading; I would suggest to add something to the title to indicate the in vitro nature and/or that the in vivo work is conducted in animal models.

Comment 4: The authors don't present any strengths or weaknesses regarding the research. I also don't find any comment on potential bias of the study or what measures were implemented to reduce the impact of bias on the study.

Reviewer #2 (Comments for the Author):

Kiliç et al show that amiodarone acts as a HDT for mycobacterial infections by promoting autophagy in macrophages. I enjoyed reading this paper and appreciate that a lot of work went into this study. I however have some methodological concerns. First, I have to note that the observed effects are quite modest (<50%) which raises my first question: Why were such short treatment windows chosen (ie 1 day in macrophages and only 4 days in zebrafishes)? Wouldn't longer treatments yield stronger results? Has this been attempted and if not, why not?

The authors also repeatedly refer to mycobacterial survival, yet based on the data it is not clear whether amiodarone is killing vs inhibiting replication. Please address.

It's also not clear to me why some experiments were done in the absence of infection (ie Fig 2J for TFEB) and some were done with infection. Additionally, why not include interferon-gamma stimulated macrophages +- AMD as a relevant control?

And the way Mmar is measured in the Zebrafish model with microscopy is not very convincing. How good is the microscopic method for detecting single Mmar cells? Why not just perform CFUs?

The pH of the phagolysosome is acidic, which restrains Mtb growth. I recommend repeating experiments Fig 1C and 1E at acidic pH (ph 5 and ph 6, though for ph 5 you will need to perform CFU since Mtb will not typically grow under standard conditions, unless you supplement with Oleic Acid [PMID: 34341117]) and in PBS as a non-acidic way to simulate non-replication.

Finally, there is clear indication of lane splicing in the Western blot in Fig 2A.

Minor comments:

Effectivity -> efficacy

Rephrase, confusing as it may be construed that Mtb is the cause of NTM -> "26 Mycobacterium tuberculosis (Mtb), the causative agent of tuberculosis (TB), as well as 27 nontuberculous mycobacteria (NTM)"

"AN in vivo" -> "a in vivo tuberculosis"

Missing word: "slowly declining [IN] the last decades" Also TB cases actually went up recently up due to the COVID19 pandemic so this sentence is misleading

Missing word -> "One 350 day prior experimental procedures, m"

Please describe the MeJuSo cell line

Please describe whether 7h9 contained ADN or OADC, and tween or tyloxapol

"3-aminobenzoid acid"?

Fig 1B: What does each circle in the bar graphs represent? And why are the DMSO circles all exactly the same value (ie 100%) - in other words why are there no error bars for DMSO?

Fig 1B is CFU but Fig 1D is MGIT? Why? Also please change the figure so it is crystal clear to the reader that one is CFU and one is MGIT. Why not get rid of the relative 100% scale and just present the actual raw data (ie CFU/mL for 1B and hours for 1D?)

"[MACROPHAGE] cell 143 death also results in reduced intracellular bacterial burden"

Fig 2D and 2G: Why doesn't Hoechst staining overlap with the LC3/Wasabi/p62 staining in some panels?

**Revision manuscript “Host-directed therapy with amiodarone in preclinical models restricts mycobacterial infection and enhances autophagy” May 2024
American Society for Microbiology – Microbiology Spectrum**

Editor comments:

1. Figure-1B, 1D and 1F- please provide the actual raw values/numbers for each group in these figures as supplementary data.

As Supplementary Figure 1, we have now included graphs showing the actual counts from the CFU assays for Mtb and the estimated CFU counts from the MGIT assay for Mav.

2. Figure 2A- please provide the raw/full length gel picture(s) as supplementary data.

The full length gel pictures used for Figure 2A are added as Supplementary Figure 2.

3. Figure 2D and G- the white arrows are barely visible; it would look better/clear if the size can be increased.

Thank you for the remark; we have increased the size of the arrows.

4. Figure-3B, and 3C- please provide the actual raw values/numbers for each group in these figures as supplementary data.

Graphs showing the actual numbers of fluorescently bacteria detected in the zebrafish are included as Supplementary Figure 4.

5. Figure-4 has 2 panels labelled as C and none with D. Kindly correct this error.

Thank you for the remark. This has been adjusted.

6. Line 208. It is important to disclose if the highest dose of AMD induced any developmental toxicity, even if it was not quantified?

AMD treatment with the highest dose, which is 20 uM, indeed resulted in measurable toxicity. With 5 or 10 uM we did not see any signs of toxicity, and for our kinetic experiments (Fig 3C) we used 5 uM to rule out any possibilities of non-observable toxicity. The text in the manuscript has been changed to disclose the remarks on toxicity:

“Amiodarone reduced bacterial burden in a dose-dependent manner at 5 and 10 μ M (Fig. 3B, Fig. S4), without showing any signs of toxicity in zebrafish embryos. The highest dose tested (i.e. 20 μ M) induced developmental toxicity (e.g., oedema and lethality) and bacterial loads were therefore not quantified.”

7. Lines 94-96. It appears that Ref#24 reports on antipsychotic drugs and no mentioning of amiodarone. Clarify and revise.

It is correct that this reference (Ref25 in the revised manuscript) focuses on antipsychotic drugs. These antipsychotic drugs were identified by screening an autophagy library (Enzo Lifesciences) on Mtb-MelJuSo cells in the study described in Ref25. This library also included amiodarone, so we refer to the efficacy of amiodarone in this screening (Line 29 of Supplementary Information 2 of Ref25). We have revised the sentence citing Ref 25 to clarify this point.

8. Line 134-135. This is confusing, since, indeed, AMD showed a direct antimycobacterial effect on M. marinum as per this paper. Kindly revise this sentence and clarify the differential responses between Mtb, Mav and Mm regarding AMD activity.

We have adjusted the sentence in lines 134-135 to describe the absence of direct antimycobacterial effect on Mtb specifically:

“To exclude direct antibacterial effects, Mtb in liquid broth was exposed to Amiodarone at the same concentration, which did not show any effect of amiodarone (Fig. 1C), thereby confirming amiodarone acts in a host-directed manner [during Mtb infection]”

9. Line 153. It is important to explain what is the rationale go after M2 macrophages and not M1 for subsequent assays?, and what is the impact of AMD on autophagy in M1 cells with or without mycobacterial infection?

In the text we only mentioned why we focused on Mav-infected cells, however, we indeed did not explain why we focused on the M2 macrophage subtype. We now elaborated the explanation:

“Considering that amiodarone showed the most consistent effect in Mav- versus Mtb-infected macrophages (standard deviation of 20.1 compared to 28.7, respectively) [and that M2 macrophages better resemble the alveolar macrophages, which are the primary cells involved during mycobacterial infections (46, 47)], we focused on Mav-infected M2 macrophages.”

Refs 46, 47: PMID: 16330536, PMID: 15070757

Furthermore, in response to your question, we have also assessed the effect of AMD on autophagy in Mav-infected M1 macrophages. Consistent with the results in M2 macrophages, in M1 macrophages we also observed that inhibition of autophagy with bafilomycin resulted in the loss of efficacy of AMD on reducing bacterial survival.

10. Based on this report, it appears that Mm and Mtb are behaving differently in response to different doses of AMD (compare Figures 1C, 1E and Suppl.Fig1A). Similarly, Mtb and Mav infected macrophages differ in the way they were handled through autophagic pathways (lines 261-270). However, the data shown in Figure 2 pertains only to Mav-infected macrophages and not Mtb-infected counterpart. Since these are major weaknesses, it is important to include a section on the limitations of this study, particularly the different model systems used, and different doses of AMD used in these models and lack of direct mechanistic link between the targeted pathways etc., at the end of discussion section.

It is correct that Mmar and Mtb are differently affected by AMD when used in a concentration of 10 μ M. In addition, in lines 261-270 we indeed refer to that Mtb can escape the phagosome, whereas this is unclear for Mav. Hence, Mtb and Mav intracellular behavior could be differently affected by autophagy and AMD treatment. We have included a paragraph in the discussion to highlight, amongst others, this limitation.

“Our study might have several limitations. First, although we identified HDT activity of amiodarone against multiple mycobacterial species, the role of autophagy in infection control by amiodarone was only shown during Mav infection. If mycobacteria differ in their intracellular behavior and the way they are degraded, our findings regarding autophagy during Mav infection might not apply to Mtb or Mmar. Second, we showed that induction of autophagy by amiodarone is required for infection control and that TFEB activation upon amiodarone treatment is enhanced. However, our study lacks evidence that TFEB activation is promoting the autophagy-mediated HDT activity of amiodarone. Third, we did not evaluate the efficacy of amiodarone in combination with standard-of-care antibiotics to detect any cumulative or synergistic effects. Clinical application of HDT, however, will most likely be considered as an adjunctive therapy to standard-of-care used to treat mycobacterial infections (11). Despite these limitations, by showing amiodarone improved host control of mycobacterial infections both in vitro and in vivo, we provide support for the possible clinical applicability of amiodarone. Furthermore, our study presents a new autophagy-inducing compound suitable for drug repurposing. Drug repurposing as HDT has various advantages, including the known safety profile of drugs and the faster facilitation of the identification of HDT to treat mycobacterial infections.”

Reviewer #1 (Comments for the Author):

Summary:

The authors identified amiodarone as a potential HDT candidate that inhibited both intracellular Mtb and Mycobacterium avium (Mav) in primary human macrophages, without directly impairing bacterial growth, thereby confirming that amiodarone acts in a host-mediated manner. They demonstrate that amiodarone induced autophagy and reduced bacterial burden in a zebrafish embryo model of TB, thereby confirming the HDT activity of amiodarone in vivo. They conclude that amiodarone is an autophagy-inducing anti-mycobacterial HDT that improves host control of mycobacterial infections.

Comment 1: Please use the latest WHO Global TB report when reporting TB figures.

We have replaced the report from 2022 with that of 2023 and adjusted the number wherever appropriate.

Comment 2: Lines 114-121: this section appears at the end of the introduction, typically where the authors would state the aim, objective, hypothesis, rationale for the research, but this section reads rather like a summary or conclusion that one would find at the end of a manuscript. I would recommend that the authors rewrite this section removing any comments on findings or conclusions made.

We do understand that a summary of our findings/conclusions at the end of the introduction is less common for research articles published in Microbiology Spectrum. We therefore replaced the end section of the introduction with the following:

“In this study, we aimed to investigate amiodarone as HDT against multiple mycobacterial species in primary human macrophages. Moreover, to understand the mechanism of action of amiodarone, we evaluated the effect of amiodarone on autophagy and the role of autophagy during infection control by amiodarone. Finally, we assessed the efficacy of amiodarone in a zebrafish TB model to determine the in vivo translatability.”

Comment 3: The title is slightly misleading; I would suggest to add something to the title to indicate the in vitro nature and/or that the in vivo work is conducted in animal models.

To cover both the cellular and animal models used in our study, we adjusted the title to:

“Host-directed therapy with amiodarone [in preclinical models] restricts mycobacterial infection and enhances autophagy”

Comment 4: The authors don't present any strengths or weaknesses regarding the research. I also don't find any comment on potential bias of the study or what measures were implemented to reduce the impact of bias on the study.

We fully agree that it is informative to add a section to the discussion on strengths and weaknesses, which we now present in the manuscript as follows:

“Our study might have several limitations. First, although we identified HDT activity of amiodarone against multiple mycobacterial species, the role of autophagy in infection control by amiodarone was only shown during Mav infection. If mycobacteria differ in their intracellular behavior and the way they are degraded, our findings regarding autophagy during Mav infection might not apply to Mtb or Mmar. Second, we showed that induction of autophagy by amiodarone is required for infection control and that TFEB activation upon amiodarone treatment is enhanced. However, our study lacks evidence that TFEB activation is promoting the autophagy-mediated HDT activity of amiodarone. Third, we did not evaluate the efficacy of amiodarone in combination with standard-of-care antibiotics to detect any cumulative or synergistic effects. Clinical application of HDT, however, will most likely be considered as an adjunctive therapy to standard-of-care used to treat mycobacterial infections (11). Despite these limitations, by showing amiodarone improved host control of mycobacterial infections both in vitro and in vivo, we provide support for the possible clinical applicability of amiodarone. Furthermore, our study presents a new autophagy-inducing compound suitable for drug repurposing. Drug repurposing as HDT has various advantages, including the known safety profile of drugs and the faster facilitation of the identification of HDT to treat mycobacterial infections.”

Reviewer #2 (Comments for the Author):

Kilinç et al show that amiodarone acts as a HDT for mycobacterial infections by promoting autophagy in macrophages. I enjoyed reading this paper and appreciate that a lot of work went into this study. I however have some methodological concerns.

1. First, I have to note that the observed effects are quite modest (<50%) which raises my first question: Why were such short treatment windows chosen (ie 1 day in macrophages and only 4 days in zebrafishes)? Wouldn't longer treatments yield stronger results? Has this been attempted and if not, why not?

It is correct that Amiodarone resulted in ~50% reduction of intracellular bacterial survival after 24 hours of treatment. We also have assessed the efficacy of amiodarone after longer treatment durations (up to 144 hours, see below), which showed equal efficacy at different time points. Longer treatment did not necessarily result in stronger effects but did increase variation between experiments and made assessment of colocalization in our confocal experiments difficult. Consequently, we focused on 24h incubation throughout the manuscript. Please note that amiodarone is a host-directed compound, and therefore its efficacy is not affected by bacterial growth speed, in contrast to classical antibiotics whose efficacy kinetics are highly depending on bacterial growth speeds.

As for the zebrafish model: the maximum duration of our experiments is limited to 4 days post infection, which equals 5 days post fertilization. Until this developmental stage zebrafish larvae are not considered as experimental animals, according to European legislation (EU Directive, 2010/63/EU)). Thus, in the light of the 3R-policy of animal experiment law, experiments must be limited to 5 days, if possible, as is the case in the Mmar-infection model.

2. The authors also repeatedly refer to mycobacterial survival, yet based on the data it is not clear whether amiodarone is killing vs inhibiting replication. Please address.

That is an interesting remark raised by the reviewer, which we did attempt to address.

Discrimination of whether bacteria are killed or the replication is inhibited would require the assessment of the metabolism of bacteria. We aimed to investigate this by generating a bioluminescent Mav strain that could indicate the intracellular metabolic activity of Mav, however, such a strain is not usable at this point. In addition, mycobacteria in stressful (intracellular) conditions may undergo dormancy, which also causes difficulty in assessing the effect of AMD on the metabolic activity of mycobacteria after treatment. Hence, at this point it is unclear whether AMD improves killing or impairs mycobacterial replication. Nevertheless, the fact that inhibition of autophagy impairs host control of infection upon AMD treatment suggests that AMD enhances autophagy-mediated killing rather than impairing intracellular growth.

We added a sentence to the results to disclose that it is unclear if its killing or inhibition of replication induced by amiodarone:

“Taken together, although we cannot discriminate between bacterial killing or restriction of replication, host-directed therapy with amiodarone controls Mav infection in primary human macrophages by promoting anti-mycobacterial autophagy, which correlated with activation of the master transcriptional regulator TFEB.”

3. It's also not clear to me why some experiments were done in the absence of infection (i.e. Fig 2J for TFEB) and some were done with infection. Additionally, why not include interferon-gamma stimulated macrophages +/- AMD as a relevant control?

Due to the absence of pictures showing fluorescent Mav Wasabi, we understand why this was not clear to the reviewer. However, also experiments in 2J were performed in the presence of an infection, as mentioned in the figure legend of figure 2J. For the clarity, we now added images of Mav-Wasabi bacteria to 2J, similarly as for Figures 2D and 2G.

Furthermore, all experiments were performed in the presence of infection. The only exception in which infection was absent, is figure 4A-B, in which we tested the effect of amiodarone on the count of LC3 clusters in zebrafish.

It would indeed be interesting to determine the role of IFNG during Mav infections in future work. We feel, however, that this falls outside the scope of the current paper. As we were interested in deciphering the mechanism of action of amiodarone, we assessed TFEB translocation in presence of Mav infection, with or without amiodarone.

4. And the way Mmar is measured in the Zebrafish model with microscopy is not very convincing. How good is the microscopic method for detecting single Mmar cells? Why not just perform CFUs?

Many studies from different laboratories have enumerated bacterial fluorescent counts/intensity with fluorescence microscopy (a few examples are: PMID: 20211140, PMID: 22453760, PMID: 24945994). This has become a widely accepted standard method in zebrafish infection studies and considered less error prone than disintegrating larvae for CFU counts. One of the major advantages of the zebrafish model is the transparency of the larvae, making it possible to assess in situ bacterial loads by quantification of fluorescent bacteria by fluorescence microscopy without any interventions. In addition, using fluorescence microscopy to enumerate bacteria also enables to follow the development of infection in the same group of embryos (which has been done for the kinetic experiments shown in Figure 3C), whereas CFU assay would otherwise have required embryo disintegration. We therefore used fluorescence microscopy to assess bacterial survival in zebrafish.

5. The pH of the phagolysosome is acidic, which restrains Mtb growth. I recommend repeating experiments Fig 1C and 1E at acidic pH (pH 5 and pH 6, though for pH 5 you will need to perform CFU since Mtb will not typically grow under standard conditions, unless you supplement with Oleic Acid [PMID: 34341117]) and in PBS as a non-acidic way to simulate non-replication.

Thank you for this suggestion. We have assessed the effect of amiodarone (10 μ M) on bacterial growth of Mav and Mtb up to 6 days in culture, in regular 7H9 (pH 6.8), acidified 7H9 (pH 6 and pH 5) and in PBS. For Mtb we indeed observed acidic conditions affects bacterial growth, with minimal growth in 7H9 pH 5 and PBS. However, we do not observe direct effects of amiodarone on Mav or Mtb growth regardless of the acidity of the medium. Hence, even in restraining conditions, amiodarone does not directly inhibit bacterial growth.

6. Finally, there is clear indication of lane splicing in the Western blot in Fig 2A.

That is correct. We have adjusted the image of Fig 2A to improve the splicing. We now added the full blot as Supplementary Figure 2. In figure 2A, representative blots for each marker are shown. Within these Western blot experiments other conditions which are not related to this manuscript were included. For this reason, these blots were spliced to visualize only the conditions (DMSO, AMD, DMSO+Baf, AMD+Baf) (blue squares) discussed in figure 2B and 2C.

7. Minor comments:

Effectivity -> efficacy

Thanks for the remark. We corrected this.

8. Rephrase, confusing as it may be construed that Mtb is the cause of NTM -> "26 Mycobacterium tuberculosis (Mtb), the causative agent of tuberculosis (TB), as well as 27 nontuberculous mycobacteria (NTM)"

We agree that this might cause confusion, so we rephrased as follows:

"Mycobacterium tuberculosis (Mtb) [the causative agent of tuberculosis (TB)], as well as nontuberculous mycobacteria (NTM) are intracellular pathogens whose treatment is extensive and increasingly impaired due to the rise of mycobacterial drug resistance."

9. "AN in vivo" -> "a in vivo tuberculosis"

Thank you for the remark. We have corrected this.

10. Missing word: "slowly declining [IN] the last decades" ☒ Also TB cases actually went up recently up due to the COVID19 pandemic so this sentence is misleading

We have adjusted the sentence and added that COVID19 likely affected this decline (reference: Cilloni et al. (2020). The potential impact of the COVID-19 pandemic on the tuberculosis epidemic a modelling analysis.):

"While the number of TB cases is slowly declining [in] the last decades, [a trend that may well be broken as a result of the COVID-19 pandemic] (ref Cilloni et al.)"

11. Missing word -> "One 350 day prior experimental procedures, m"

We have adjusted the sentence.:

"One day prior [to] experimental procedures"

12. Please describe the MeJJuSo cell line

The rationale for using MeJJuSo has been stated before: 'Korbee, C. J et al. Combined chemical genetics and data-driven bioinformatics approach identifies receptor tyrosine kinase inhibitors as host-directed antimicrobials', which is now added as reference. However, as the MeJJuSo cell line was not part of our current study, we removed mentioning this cell line and changed these particular sentences to:

"During an in vitro screen of a library composed of autophagy modulating compounds using a human the MeJJuSo cell line infected with Mtb, we have previously identified amiodarone to reduce bacterial burden."

"To identify new drugs with host-directed therapeutic (HDT) activity against intracellular Mtb, we have previously screened the Screen-Well autophagy library of clinically approved molecules by treating Mtb-infected human MeJJuSo cells for 24 hours."

13. Please describe whether 7h9 contained ADN or OADC, and tween or tyloxapol

The 7H9 used for bacterial cultures contains tween, glycerol and ADC, so we added this information:

"Mtb, Mav, and Mmar cultures were diluted to an OD600 of 0.1 in [Difco Middlebrook] 7H9 broth [(Becton Dickinson, Breda, the Netherlands), containing 0.2% glycerol (Merck Life Science, Amsterdam, The Netherlands), 0.05% Tween-80 (Merck Life Science), 10% Middlebrook albumin, dextrose, and catalase (ADC) enrichment (Becton Dickinson), and 100 µg/mL Hygromycin B (Life Technologies-Invitrogen, Bleiswijk, The Netherlands)]...."

14. "3-aminobenzoid acid"?

We have corrected the name to "*3-aminobenzoic acid ethyl ester*".

15. Fig 1B: What does each circle in the bar graphs represent? And why are the DMSO circles all exactly the same value (ie 100%) - in other words why are there no error bars for DMSO?

As the legend of Figure 1B describes: "*Dots represent the mean from triplicate wells of a single donor*".

We normalized the values for each technical replicate, including that of the control, to the average of the control (DMSO) within each donor. When the average of the technical replicates of the control is taken (to end up with one value per donor), this will be 100%.

For the clarity, we now added the information that normalization was performed per donor for the normalized data, as follows:

Legend figure 1B: "*Dots represent the mean from triplicate wells of a single donor. Bacterial survival is expressed as percentage of vehicle control DMSO [(=100%, indicated with the dotted line) per donor]. Statistical significance was tested using a paired t-test.*"

16. Fig 1B is CFU but Fig 1D is MGIT? Why? Also please change the figure so it is crystal clear to the reader that one is CFU and one is MGIT. Why not get rid of the relative 100% scale and just present the actual raw data (ie CFU/mL for 1B and hours for 1D?)

In the figure legends of 1B and 1D the specific assay used (CFU or MGIT) is mentioned. We now have adjusted the y-axis of these figures, so it is also clear from the figure that the data in Figure 1B derived from CFU assays and the data in Figure 1D are the estimated CFU counts based on the MGIT assay (see below). To answer your question why we used MGIT assay for the Mav infection experiments: by showing a strong correlation with results from the CFU assay, we previously validated the MGIT assay as a valid alternative to the CFU assay to assess intracellular bacterial load during Mav infection (PMID: 35811670). Moreover, the MGIT assay has higher objectivity, and we also showed less variation compared to the CFU assay.

We decided to show the normalized data because the intracellular bacterial loads sometimes do differ between donors, but most importantly the effect size of AMD compared to control maintains similar among donors which the latter we wanted to emphasize by showing normalized data. Also upon request by the editor, we included the raw data of Fig 1B and 1D as Supplementary Figure 1.

17. "[MACROPHAGE] cell 143 death also results in reduced intracellular bacterial burden"

We can understand that "cell" in this context may also imply the bacterial cell, so we have included your suggestion for clarification.

".....[macrophage] cell death also results in reduced intracellular bacterial burden."

18. Fig 2D and 2G: Why doesn't Hoechst staining overlap with the LC3/Wasabi/p62 staining in some panels?

Hoechst is a nuclear stain that we do not expect to necessarily overlap with autophagy related markers, or bacteria, but may however get into close vicinity. Please note in our confocal experiments, we have not collected z-planes, so the LC3/Wasabi/p62 may appear to be within the nucleus, but in fact is in another imaging plane.

Re: Spectrum00167-24R1 (Host-directed therapy with amiodarone in preclinical models restricts mycobacterial infection and enhances autophagy)

Dear Dr. Anno Saris:

Your manuscript has been accepted, and I am forwarding it to the ASM production staff for publication. Your paper will first be checked to make sure all elements meet the technical requirements. ASM staff will contact you if anything needs to be revised before copyediting and production can begin. Otherwise, you will be notified when your proofs are ready to be viewed.

Sincerely,
Selvakumar Subbian
Editor
Microbiology Spectrum